# Advancing Physiological Time Series Reconstruction and Imputation via Mixture of Receptive Fields and Experts Fusion

## Abstract

Recent studies show that using diffusion models for time series signal reconstruction holds great promise. However, such approaches remain largely unexplored in the domain of medical time series. The unique characteristics of the physiological time series signals, such as multivariate, high temporal variability, highly noisy, and artifact-prone, make deep learning-based approaches still challenging for tasks such as imputation. Hence, we propose a novel Mixture of Experts (MoE)-based noise estimator within a score-based diffusion framework. Specifically, the Receptive Field Adaptive MoE (RFAMoE) module is designed to enable each channel to adaptively select desired receptive fields throughout the diffusion process. Moreover, recent literature has found that when generating a physiological signal, performing multiple inferences and averaging the reconstructed signals can effectively reduce reconstruction errors, but at the cost of significant computational and latency overhead. We design a Fusion MoE module and innovatively leverage the nature of MoE module to generate $K$ noise signals in parallel, fuse them using a routing mechanism, and complete signal reconstruction in a single inference step. This design not only improves performance over previous methods but also eliminates the substantial computational cost and latency associated with multiple inference processes. Extensive results demonstrate that our proposed framework consistently outperforms diffusion-based SOTA works on different tasks and datasets.

## 1 Introduction

Medical time series data have gained increasing attention with the advancement of deep neural networks. Among these, physiological time series signals, such as Electrocardiogram (ECG), Electroencephalogram (EEG), and Polysomnography (PSG), are particularly important as they directly reflect human physiological states, driving progress in areas such as disease diagnosis, patient monitoring, and treatment outcome prediction. Recent studies have explored various tasks, such as reconstruction Li et al. (2023a) and imputation Lim & Zohren (2021), to address issues of data incompleteness and noise. For example, ECG reconstruction aims to restore corrupted or missing signal segments caused by uncontrollable factors, ensuring the reliability and continuity of ECG data, which is essential for accurate clinical interpretation and decision-making.

Recently, score-based diffusion models have achieved state-of-the-art performance in time series reconstruction compared to traditional methods Wang et al. (2023); Alcaraz & Strodthoff (2022); Tashiro et al. (2021). The success in time series reconstruction can be attributed to their abilities to model complex data distributions, generating realistic outputs through iterative refinement, and automatic feature extraction in conjunction with inductive biases.

However, physiological time series signals differ significantly from common time series data. They often lack clear long-term periodic patterns. Additionally, as each sample is collected from different patients, the periodicity for the same channel across different samples exhibits substantial variability. Besides, different channels within the same sample are collected by different sensors placed on different parts of the human body, leading to highly inconsistent periodicity across channels within a single training sample, as exemplified by ECG and PSG signals, which exhibit complex patterns and distinctive physiological characteristics.

Hence, a single DNN model commonly struggles to achieve low reconstruction errors across all channels within the same sample and fails to maintain consistent performance across the same channels in different samples. Fig. 1 illustrates the limitations of the single model Li et al. (2023a) in handling the physiological time series dataset, as it struggles to maintain consistent performance across all channels within a single sample and fails to ensure stable performance across the same channel in different samples.

Mixture of Experts (MoE) has achieved remarkable success in the fields of computer vision (CV) and natural language processing (NLP). This success is primarily attributed to MoE's ability to effectively specialize each expert model, enabling different expert models to extract distinct features from the data. These features are then combined through a weighted summation to produce the final output, thereby maximizing the model's performance. Meanwhile, sparsely-gated mixture-of-experts Shazeer et al. (2017) has emerged as an effective strategy for selectively activating a subset of experts based on input data, guided by a router. This mechanism allows the network to significantly enhance its capacity without incurring a proportional increase in computational cost.

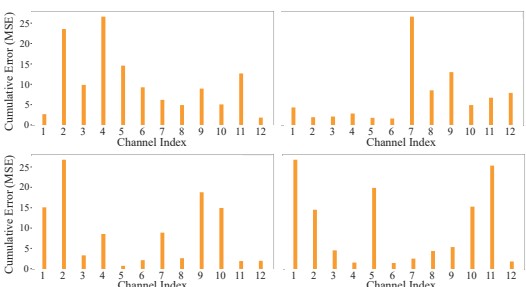

Figure 1: The demonstration of the inconsistency in reconstruction quality among 12 different channels within a data sample and 4 different data samples. The data samples are the 12-channel ECG signals that are randomly selected from PTB-XL dataset and reconstructed by the SOTA method DeScoD-ECG Li et al. (2023a).

Traditional MoE models typically utilize identical experts across data samples. However, in the context of ECG signals, this pattern is less suitable due to the sensitivity of ECG and PSG signals to receptive fields. Since MoE has not previously been applied to physiological signal processing, it remains unclear which parts of the network can most effectively be replaced by the MoE layer. To bridge this gap, we propose the Receptive Field Adaptive Mixture of Experts (RFAMoE) block, where we deploy different sizes of convolution kernels to enable each channel in the feature map to independently select its proper convolution kernel size. It prevents channels with different preferred receptive fields from being constrained to use the same kernel size. We found that this method can effectively solve the above two challenges.

In addition, the previous study Li et al. (2023a) has proven that performing multiple reconstructions of the physiological time series signals using the same model and averaging the results (K-shot Averaging) can effectively improve the accuracy of the reconstruction. However, this approach significantly increases the computational burden due to repeated inference, making it impractical for use in real-world medical applications. Based on this finding, and to avoid the significant computational burden, we innovatively leverage the nature of MoE to simultaneously generate multiple denoising factors for each channel at the output of the backbone, then utilize the fused denoising factor via router gates to reconstruct the signals in a single inference computation. Our observations show that our proposed fusion MoE shares a similar principle with the method that requires averaging multiple reconstructions, but our fusion MoE is much efficient. Furthermore, we theoretically prove that our fusion MoE outperforms averaging multiple reconstruction results in reducing the errors of reconstruction.

## 2 RELATED WORKS

### 2.1 DIFFUSION MODEL

Diffusion models have recently gained significant attention in time series modeling, particularly for reconstruction, imputation, forecasting, and anomaly detection. Tashiro et al. (2023) introduced TimeDiff, which demonstrated superior performance in time series forecasting and classification. Kollovieh et al. (2023) proposed TSDiff, integrating self-guidance mechanisms to refine generative outputs, making it applicable to both unconditional and conditional time series synthesis. A broader survey by Lin et al. (2024) categorized diffusion-based approaches in time series and spatio-temporal data.

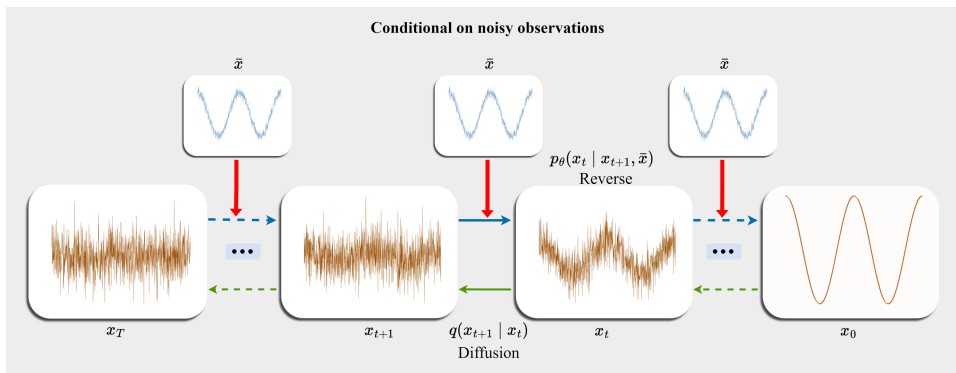

Figure 2: Overview of conditional diffusion for time series reconstruction.

However, despite the rapid advancements in general time series modeling, diffusion-based approaches have not been extensively explored in the physiological time series domain. Some recent works have applied diffusion models for EHR data generation Li et al. (2023b) and physiological signal imputation Wang et al. (2022). Additionally, Wang et al. (2024) investigated their use in multi-modal medical applications, including physiological signals and medical imaging. Furthermore, diffusion-based models have been employed for time series anomaly detection Shen et al. (2023) and denoising in ICU patient monitoring Zhang et al. (2023). Despite these efforts, the development of diffusion models specifically tailored for physiological time series reconstruction remains limited.

This gap highlights the need for further exploration of diffusion-based models in medical time series reconstruction. Our work aims to address this challenge by leveraging diffusion models to improve the quality and reliability of medical time series signal reconstruction, which is crucial for downstream clinical applications.

## 2.2 MIXTURE OF EXPERTS

Mixture of Experts (MoE) models have been widely adopted in tasks involving heterogeneous data due to their ability to dynamically route inputs to specialized sub-models, effectively handling diverse data distributions. For instance, Shazeer et al. (2017) introduced an MoE layer within a neural network to improve language modeling tasks, demonstrating significant performance gains. Similarly, Riquelme et al. (2021) applied MoE architectures to computer vision, achieving state-of-the-art results in image classification.

In the realm of time series forecasting, Shi et al. (2024) proposed Time-MoE, a scalable architecture designed to pre-train large forecasting models while reducing inference costs. This model leverages a sparse MoE design to enhance computational efficiency. Additionally, Liu et al. (2024) introduced Moirai-MoE, the first MoE time series foundation model, achieving token-level model specialization in a data-driven manner.

Despite these advancements, the application of MoE models in the medical time series domain remains underexplored. Notably, Lee & Hauskrecht (2022) proposed a residual MoE framework to adapt clinical sequences, highlighting the potential of MoE in this area. However, comprehensive studies leveraging MoE for medical time series reconstruction are scarce. Given the high heterogeneity across channels and significant intra-population variability in physiological time series data, in addition to the lack of dedicated MoE solutions, we propose a novel diffusion-based MoE framework specifically designed for reconstructing and imputing physiological time series signals. Our approach leverages diffusion-based generative modeling to enhance temporal consistency and adaptive expert selection to dynamically handle diverse signal characteristics, ensuring robust and accurate reconstructions across different patient populations. And the framework of the condition diffusion model used in this paper is shown in Fig 2, where $\bar{x}$ is the condition for each time step that guides the model to reconstruct signals.

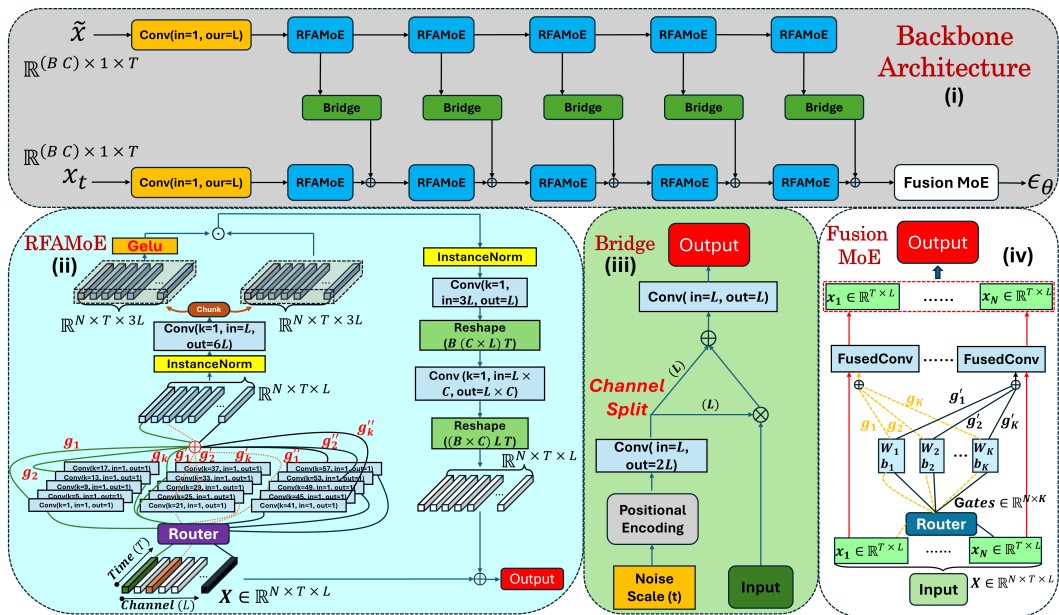

Figure 3: The framework of our method: (i) The backbone architecture of the whole framework, (ii) RFAMoE block: to handle the multi-variate adaptive fields on different channels and samples. (iii) The bridge block: to affine the feature map extracted on conditions to the certain space according to the current time step ($t$ represents the time step, noise scale is set by time steps), (iv) Fusion MoE: to generate fused denoise factor to further improve accuracy of reconstruction of signals.

## 3 METHOD

In this section, we introduce three key components for the backbone, followed by the overall structure of the backbone. First, we introduce the structure of RFAMoE block and Bridge block shown as (ii) and (iii) in Fig.3 and the function of their components. Second, we introduce the fusion MoE as (iv) in Fig 3 and prove that fusion MoE results in smaller errors compared to K-shot averaging inside the each step reconstruction. And it empirically shares the similar principle with K-shot averaging on reconstructing signals. Third, we introduce the structural design of backbone depicted as (i) in Fig. 3.

### 3.1 RFAMoE AND BRIDGE

The input to the RFAMoE block consists of $N$ feature maps, where $N = B \times C$. Here, $B$ represents the batch size, and $C$ denotes the number of channels in each sample. $T$ denotes the time length of the time series signals, and $L$ represents the number of channels in the feature maps. The RFAMoE block utilizes a combination of multi-scale convolution layers to extract features from channels with different receptive fields. The dimension $N$ encompasses all channels of all samples. When using the multi-scale convolutional expert group to process $N$ feature maps, each channel within the feature maps can independently select the most suitable kernel size, rather than enforcing a uniform kernel size with the same receptive field across channels at the same position within the $N$ feature maps Wang et al. (2025); Li et al. (2023a); Romero et al. (2021), which particularly disadvantageous for medical time series data that are highly sensitive to receptive fields. $k$ is the number of convolutional layers activated by each channel, referred to as top-$k$ policy. These extracted features are then weighted and aggregated based on weights assigned by the router, resulting in the final feature representation for each channel. In our method, we found that top-1 activation serves as the optimal candidate activation strategy for our datasets.

Following the module of multi-scale convolution expert layers, the instance normalization is applied to each channel, which facilitating convergence and maintain the statistical independence of features, ensuring they are not affected by inter-channel interference. The feature map after instance normalization is divided into two parts. One part undergoes a GELU activation, followed by element-wise multiplication with the other part Ma et al. (2024), explicitly increasing the network width. At the end of the RFAMoE block, the feature map $\mathbf{X} \in \mathbb{R}^{N \times T \times L}$ is reshaped to $\mathbf{X} \in \mathbb{R}^{B \times (C \times L) \times T}$, then

---

**Algorithm 1:** Dynamic Convolution in Fusion MoE

---

**Require:** Input feature maps $\boldsymbol{X} \in \mathbb{R}^{N \times T \times L}$, experts with weights and biases, router for gating.
**Ensure:** Processed feature maps $\boldsymbol{Y} \in \mathbb{R}^{N \times T \times 1}$.
 1: **Step 1: Compute Gates.**
 2: $\boldsymbol{Gates} \leftarrow$ Router($\boldsymbol{X}$). # The shape of Gates is $N \times K$, $K$ is the number of experts.
 3: **Step 2: Fuse expert weights using gates.**
 4: $weights \leftarrow$ Stack(Weights of Experts). # Shape: $(K, 1, L, S)$, $S$ is the kernel size
 5: $merged\_weights \leftarrow$ Einsum($\boldsymbol{Gates}, weights$). # Shape: $(N, 1, L, S)$
 6: **Step 3: Dynamic convolution for each sample.**
 7: Initialize $outputs \leftarrow []$.
 8: **for** $i = 1$ **to** $N$ **do**
 9: $\quad weight \leftarrow merged\_weights[i]$. # Weight for sample $i$.
10: $\quad output \leftarrow$ Conv1D($\boldsymbol{X}[i], weight$). # Dynamic convolution.
11: $\quad$ Append $output$ to $outputs$.
12: **end for**
13: $\boldsymbol{Y} \leftarrow$ Concat($outputs, \text{axis} = 0$)
14: # Shape: $(N, T, 1)$
15: **Return processed feature maps: $\boldsymbol{Y}$.**

---

the information interaction is performed across all channels along the batch dimension. The kernel size of the convolutional layer used for channel information fusion is set to 1, ensuring the most effective fusion of inter-channel information while preserving the long-sequence feature paradigm already established within each channel Donghao & Xue (2024). The final result of the information interaction is combined with the input through a residual connection to produce the output of the block.

The Bridge block is modified based on the feature-wise linear modulation (FiLM) Dumoulin et al. (2018). Each Bridge Block projects the received feature map into a specific feature space through an affine transformation, conditioned on the noise level of the current time step.

## 3.2 Fusion MoE

Inspired by the K-shot averaging proposed by DeScoD-ECG Li et al. (2023a), and to avoid the substantial computational cost brought by K-shot averaging, we employ a fused MoE approach, enabling the network to generate a fused denoise factor in a single forward pass. Then reconstruct the signal using the fused denoise factor. In the Fusion MoE, the router computes gate weights for $N$ feature maps corresponding to $K$ expert convolution layers, each with a kernel size of 1, an input channel size of $L$, and an output channel size of 1. These gate weights are then used to fuse the weights and biases of the $K$ convolution layers into a single convolution layer. The fused convolution layer subsequently outputs the results for the corresponding feature map. The proposed framework of Fusion MoE is presented in Algorithm 1.

We prove that utilizing proposed Fusion MoE to predict multiple denoise factor, followed by fusion to reconstruct the signal, achieves better performance than K-shot averaging inside each step on reconstructing signals in theorem 3.1. The detailed proofs are provided in Appendix E.

Besides, we empirically found that Fusion MoE and K-shot averaging share a similar underlying principle in reducing reconstruction error in section C.

**Theorem 3.1** (MoE Is No Worse Than $K$-Run Averaging). *Let $\{g_\theta^{(k)}\}_{k=1}^K$ be $K$ "expert" noise estimators, each mapping $(x_t, t, \tilde{x}) \mapsto \mathbb{R}^d$. Assume there is a linear (or affine) diffusion-update operator*

$$\epsilon_D : \ \mathbb{R}^d \times \mathbb{R}^d \times \{1, \ldots, T\} \ \rightarrow \ \mathbb{R}^d \tag{1}$$

*defined by*

$$x_{t-1} = \epsilon_D(x_t, \hat{\epsilon}, t) = A(t)\, x_t + B(t)\, \hat{\epsilon}, \tag{2}$$

*where $A(t), B(t)$ are fixed linear/affine transformations that do not depend on $\hat{\epsilon}$. We compare two ways of combining these $K$ experts to produce a final $x_{t-1}$:*

1. $K$-**Run + Averaging:**

$$x_{t-1}^{(k)} = \epsilon_D\big(x_t,\, g_\theta^{(k)}(x_t,\, t,\, \tilde{x}),\, t\big),$$

$$\overline{x}_{t-1} = \frac{1}{K} \sum_{k=1}^{K} x_{t-1}^{(k)}. \tag{3}$$

2. **Single-Step Mixture-of-Experts (MoE):**

$$\hat{\epsilon}_{\text{MoE}} = \sum_{k=1}^{K} w^{(k)}(\tilde{x})\, g_\theta^{(k)}(x_t,\, t,\, \tilde{x}),$$

$$x_{t-1}^{\text{MoE}} = \epsilon_D\big(x_t,\, \hat{\epsilon}_{\text{MoE}},\, t\big). \tag{4}$$

*Here, $w^{(k)}(\tilde{x}) \geq 0$ and $\sum_{k=1}^{K} w^{(k)}(\tilde{x}) = 1$.*

*Let $L : \mathbb{R}^d \to \mathbb{R}$ be a **convex** loss function (e.g. MSE). We claim that*

$$\begin{aligned} L\big(x_{t-1}^{\text{MoE}}\big) &\leq L\big(\overline{x}_{t-1}\big), \quad \text{and hence} \\ \mathbb{E}\Big[L\big(x_{t-1}^{\text{MoE}}\big)\Big] &\leq \mathbb{E}\Big[L\big(\overline{x}_{t-1}\big)\Big], \end{aligned} \tag{5}$$

*where $\mathbb{E}[\cdot]$ is taken over all randomness (including the distribution of noise predictions $g_\theta$). Thus, a single-step MoE approach* cannot be worse *than running $K$ separate passes and averaging.*

### 3.3 NETWORK ARCHITECTURE

We adopt the backbone structure from Li et al. (2023a), with the primary difference being the deployment of the Fusion MoE module at the end of the network and RFAMoE block.

The network has two inputs: $\bar{x}$ and $x_t$. The input $\bar{x}$ is the result of masking the original signal $x$, which is used as a condition to guide the reconstruction direction of the model. In real-world medical time series data, frame loss and segment missing frequently occur due to uncontrollable factors during data acquisition. Therefore, we use masking to obtain $\bar{x}$ to simulate the time series signals obtained in real-world scenarios, making the model more adaptable to practical applications. In our experiments, we adopt an imputation strategy of masking. During training, $x_t$ is generated by scaling the original signal $x$ according to the current time step and adding Gaussian noise of a corresponding level. During inference, the first $x_t$ is replaced with random Gaussian noise and then is reconstructed step by step.

The network utilizes the RFAMoE block to hierarchically extract features from both $x_t$ and $\bar{x}$. Each hierarchical feature of $\bar{x}$ is assigned an independent bridge block, which projects the corresponding feature into a specific space based on the time step information. The projected hierarchical features are then added to the features extracted from $x_t$. Finally, a Fusion MoE is deployed at the end of the network to generate a fusion of multiple denoise factors, further improving the reconstruction accuracy.

## 4 EXPERIMENTS

**Experimental setup and baselines.** We validate the superiority of our method by comparing it against three mainstream SOTA condition-based diffusion models on imputing medical time series data (SSSDS4 Alcaraz & Strodthoff (2022); Diffwave Kong et al. (2020), DeSco-ECG Li et al. (2023a)) for imputing ECG and PSG signals. In our experiments, both our method and DeSco-ECG Li et al. (2023a) were configured with a time step of 40 and a batch size of 6, while the number of training epochs was set to 120 for both our method and DeSco-ECG. We set SSSDS4 and Diffwave as their default settings, with a time step of 200, 100 training epochs, and a batch size of 128.

All reported times in this paper represent the inference time required to reconstruct a single complete sample on a single Nvidia A6000 GPU. In our method, the parameter $L$ is set to 160. In the Fusion MoE module, we note that increasing $K$ beyond 16 yields diminishing returns, as further discussed in the D.2 of Appendix D. The number of experts $K$ is set to 16, utilizing a top-16 activation strategy, while RFAMoE employs a total of 15 convolution layer experts, utilizing the top-1 activation strategy. We compare our method with baseline methods using three similarity metrics (PRD, SSD, MAD) that quantify the quality of ECG and PSG signals reconstruction. Our accuracy results reported are the mean accuracy over 3 runs using different random seeds. In addition, we also statistic each model's parameters, FLOPs, and reconstruction steps to provide a more comprehensive comparison.

**Datasets.** In our study, we focus on physiological time-series reconstruction and imputation, which involves handling missing data in physiological signals. We selected the **PTB-XL** Wagner et al. (2020) and **SleepEDF** Kemp et al. (2000) datasets because they provide diverse, real-world time-series data across different physiological domains—cardiology (ECG) and sleep monitoring (PSG). The combination of these datasets allows us to evaluate performance in distinct but clinically relevant contexts. More details about the datasets are in the Appendix A.

**Reconstruction and imputation scenario.** We evaluate our method in two scenarios (i.e., signal reconstruction and imputation) by employing two types of signal masking strategies to simulate missing data in physiological time series: a *random mask* and a *continuous mask*. The random mask introduces missing values at arbitrary locations and of varying sizes throughout the sequence, mimicking unpredictable data loss commonly caused by sensor malfunctions, transmission errors, or manual entry omissions. In contrast, the continuous mask removes a contiguous segment of data with a fixed length, representing more structured missingness such as prolonged sensor disconnection, scheduled measurement gaps, or patient movement artifacts. These two masking approaches are designed to reflect realistic missing data patterns in clinical settings and provide a comprehensive evaluation of imputation methods under both irregular and structured missingness conditions.

### 4.1 COMPARISON TO BASELINES

Table 1 presents a comparison between our method and the baseline methods on the PTB dataset. Our method significantly outperforms the baselines in terms of the PRD metric, while also achieving superior results on SSD and MAD. Notably, the PRD metric is the most critical indicator in physiological time-series reconstruction tasks. PRD penalizes prediction errors more heavily for low-variability signals. Since medical physiological signals typically exhibit low variability, PRD plays a crucial role in diagnostic applications. From Table 2, we can also observe the same trend that our method outperforms the baseline methods in terms of the PRD metric on SleepEDF dataset. And Table 3 shows the imputation results on the PTB dataset. Though we mainly focus on diffusion-based methods in this work, we still provide SOTA non-diffusion-based methods (e.g., iTransformer, TimesNet, TimeMixer, and PatchTST) as references. More results and discussions are provided in Appendix B.

Additionally, we visualize the reconstruction results of our method with the SOTA DeScoD-ECG in Fig. 4. In the visualization, the black line represents the ground truth, while our method's output is shown in red, and the DeScoD-ECG's results are in blue. The zoomed-in area clearly tells that our method perfectly aligns the ground truth in channels exhibiting distinct periodic characteristics. More importantly, in channels with complex modalities without obvious periodicity, our method shows superior performance by better aligning the ground truth around abrupt change points, achieving higher reconstruction fidelity than the current state-of-the-art approach.

Table 1: Comparison results of reconstruction task on PTB dataset.

| | PRD ↓ | SSD ↓ | MAD ↓ | Params | FLOPs | Steps | Inference Time |
|---|---|---|---|---|---|---|---|
| iTransformer | 76.81 | 496.26 | 4.20 | 1.44M | 0.01G | NA | 0.003s |
| TimesNet | 37.43 | 144.70 | 2.45 | 4.69M | 25.01G | NA | 0.02s |
| TimeMixer | 53.14 | 243.08 | 4.07 | 9.01M | 12.70G | NA | 0.003s |
| PatchTST | 44.40 | 165.89 | 3.65 | 17.19M | 1.96G | NA | 0.003s |
| SSSDS4Imputer | 23.69 | 76.93 | 2.23 | 24.63M | 7.82T | 200 | 46.17s |
| Diffwave | 28.23 | 105.89 | 2.84 | 24.56M | 7.76T | 200 | 4.64s |
| DeScoD-ECG | 16.54 | 52.71 | 0.98 | 38.47M | 5.54T | 40 | 0.46s |
| Ours | 7.21 | 21.63 | 0.66 | 40.29M | 5.91T | 40 | 0.85s |

Table 2: Comparison results of the reconstruction task on SleepEDF dataset.

| | PRD ↓ | SSD ↓ | MAD ↓ | Params | FLOPs | Steps | Inference Time |
|---|---|---|---|---|---|---|---|
| iTransformer | 47.61 | 689.01 | 1.97 | 1.95M | 0.02G | NA | 0.003s |
| TimesNet | 39.62 | 479.99 | 2.27 | 4.69M | 76.31G | NA | 0.024s |
| TimeMixer | 28.91 | 293.73 | 1.67 | 72.04M | 55.91G | NA | 0.003s |
| PatchTST | 32.50 | 352.81 | 1.68 | 145.19M | 3.52G | NA | 0.003s |
| SSS4Imputer | 37.04 | 641.46 | 2.05 | 24.41M | 23.18T | 200 | 80.07s |
| Diffwave | 41.58 | 693.33 | 2.51 | 24.33M | 22.99T | 200 | 5.2s |
| DeScoD-ECG | 34.47 | 627.31 | 1.97 | 12.10M | 6.25T | 40 | 0.64s |
| Ours | 29.59 | 621.26 | 1.91 | 12.63M | 6.66T | 40 | 1.00s |

Table 3: Comparison results of the imputation task on PTB dataset. The 300/500/800 represents the length of the contiguously missed data from different numbers of channels (i.e., 1,3,5,7,9, and 12).

| Models Metrics | | Ours | | | DeScoD-ECG | | | iTransformer | | | TimesNet | | | TimeMixer | | | PatchTsT | | |
|---|---|---|---|---|---|---|---|---|---|---|---|---|---|---|---|---|---|---|---|
| | | PRD | MAD | SSD | PRD | MAD | SSD | PRD | MAD | SSD | PRD | MAD | SSD | PRD | MAD | SSD | PRD | MAD | SSD |
| Drop 1 Channel | 300 | **4.16** | **0.01** | **0.48** | 10.95 | 0.02 | 4.11 | 94.29 | 0.37 | 225.07 | 27.37 | 0.08 | 27.05 | 96.76 | 0.37 | 234.21 | 99.85 | 0.38 | 254.65 |
| | 500 | **4.34** | **0.01** | **0.86** | 11.61 | 0.02 | 6.88 | 96.22 | 0.38 | 384.71 | 30.77 | 0.09 | 53.11 | 100.75 | 0.39 | 418.73 | 101.95 | 0.39 | 438.46 |
| | 800 | **11.73** | **0.04** | **10.45** | 21.95 | 0.08 | 40.60 | 98.40 | 0.40 | 656.99 | 39.77 | 0.10 | 159.85 | 109.13 | 0.41 | 825.56 | 107.33 | 0.40 | 805.01 |
| Drop 3 Channels | 300 | **5.86** | **0.06** | **1.86** | 12.32 | 0.07 | 4.89 | 94.47 | 1.05 | 231.65 | 26.99 | 0.23 | 19.94 | 97.28 | 1.05 | 243.89 | 99.74 | 1.09 | 261.68 |
| | 500 | **4.65** | **0.02** | **1.53** | 12.83 | 0.08 | 12.48 | 96.59 | 1.11 | 411.96 | 30.32 | 0.26 | 54.04 | 101.34 | 1.13 | 453.94 | 101.90 | 1.13 | 469.06 |
| | 800 | **14.35** | **0.17** | **20.35** | 24.75 | 0.26 | 64.69 | 98.65 | 1.16 | 694.36 | 44.22 | 0.33 | 205.57 | 112.49 | 1.20 | 941.44 | 110.82 | 1.18 | 915.68 |
| Drop 5 Channels | 300 | **5.85** | **0.06** | **1.86** | 13.97 | 0.16 | 7.97 | 95.61 | 1.78 | 238.10 | 38.17 | 0.55 | 40.85 | 97.24 | 1.79 | 241.19 | 99.72 | 1.83 | 261.82 |
| | 500 | **19.51** | **0.36** | **18.07** | 27.88 | 0.54 | 39.28 | 97.52 | 1.88 | 416.54 | 51.89 | 0.81 | 126.46 | 101.39 | 1.89 | 442.45 | 102.06 | 1.91 | 460.62 |
| | 800 | **61.73** | 1.28 | **264.73** | 67.61 | 1.29 | 276.84 | 99.10 | 1.95 | 696.60 | 69.50 | **0.97** | 382.93 | 112.45 | 2.02 | 926.81 | 110.66 | 1.99 | 899.74 |
| Drop 7 Channels | 300 | **12.36** | **0.24** | **12.54** | 20.66 | 0.40 | 22.24 | 95.76 | 2.44 | 240.36 | 46.19 | 0.98 | 65.70 | 96.87 | 2.43 | 241.66 | 99.60 | 2.51 | 262.95 |
| | 500 | **40.02** | **1.21** | **83.25** | 47.50 | 1.39 | 115.23 | 97.82 | 2.58 | 421.45 | 66.66 | 1.64 | 228.52 | 101.17 | 2.59 | 453.29 | 102.03 | 2.61 | 468.57 |
| | 800 | **83.53** | 2.43 | **514.63** | 86.81 | 2.36 | 530.65 | 99.28 | 2.68 | 706.13 | 88.29 | **1.96** | 653.61 | 112.31 | 2.77 | 956.13 | 110.78 | 2.73 | 938.78 |
| Drop 9 Channels | 300 | **14.41** | **0.35** | **13.40** | 22.78 | 0.56 | 24.22 | 95.68 | 3.11 | 241.92 | 49.41 | 1.36 | 77.88 | 95.97 | 3.09 | 241.40 | 99.56 | 3.19 | 267.28 |
| | 500 | **37.91** | **1.48** | **84.26** | 45.88 | 1.72 | 116.61 | 97.97 | 3.28 | 430.13 | 69.04 | 2.28 | 241.68 | 100.15 | 3.28 | 445.17 | 101.77 | 3.33 | 474.70 |
| | 800 | **84.02** | 3.13 | **533.00** | 88.63 | 3.00 | 562.41 | 99.43 | 3.40 | 720.60 | 90.25 | **2.67** | 692.87 | 111.23 | 3.51 | 963.65 | 109.90 | 3.46 | 947.92 |
| Drop 12 Channels | 300 | **17.39** | **0.57** | **27.45** | 25.78 | 0.85 | 40.62 | 96.15 | 4.23 | 250.32 | 52.37 | 2.03 | 102.13 | 96.48 | 4.21 | 247.63 | 99.58 | 4.33 | 269.79 |
| | 500 | **55.88** | **3.87** | **159.95** | 60.85 | 3.97 | 186.92 | 98.54 | 4.46 | 437.44 | 78.97 | 4.09 | 314.76 | 100.36 | 4.46 | 454.00 | 101.66 | 4.50 | 470.32 |
| | 800 | **95.21** | 4.63 | **643.14** | 97.73 | 4.63 | 674.01 | 99.74 | **4.62** | 719.97 | 104.95 | 4.63 | 892.67 | 110.96 | 4.75 | 969.00 | 109.51 | 4.68 | 960.36 |

## 4.2 DeScoD-ECG K-shot vs Our Single-shot

Following the K-shot average method suggested in DeScoD-ECG, we conduct 1-shot reconstruction, 6-shot averaged reconstruction, and 12-shot averaged reconstruction on the PTB dataset using DeScoD-ECG. Then, we compared the performance of our method (1-shot) against these three K-shot settings across three evaluation metrics in Table 4, as well as the reconstruction time and computational cost for a single signal under each method. We observe that compared to K-shot averaging, our method still outperforms the 12-shot DeScoD-ECG, when its FLOPs is already 66.48T while ours is 5.91T. At this point, the inference time of 12-shot DeScoD-ECG is already six times ours. When DeScoD-ECG has a similar inference time of 0.95s as our 0.85s, its PRD performance is much worse than ours (14.44 vs 7.21). It is evident our method offers better performance while maintaining a substantially lower computational cost.

To further explore the rationale behind this superior performance, we observe the distribution of reconstruction error from DeScoD-ECG over 12 shots and compare it with the distribution of

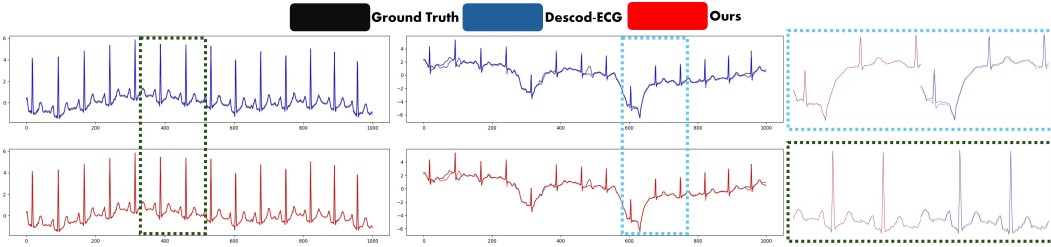

Figure 4: The reconstruction results of our method and DeScoD-ECG, where the black line represents the ground truth, the blue line corresponds to the reconstruction results of DeScoD-ECG, and the red line corresponds to our method.

Table 4: Comparison of our method with different K-shot reconstruction settings on the PTB dataset. Lower values indicate better performance.

|  | PRD ↓ | SSD ↓ | MAD ↓ | Time ↓ | FLOPs ↓ |
|---|---|---|---|---|---|
| 1-shot DeScoD-ECG | 16.54 | 52.71 | 0.98 | 0.46s | 5.54T |
| 2-shot DeScoD-ECG | 14.44 | 44.67 | 0.79 | 0.95s | 11.08T |
| 8-shot DeScoD-ECG | 11.88 | 26.79 | 0.70 | 2.83s | 44.32T |
| 12-shot DeScoD-ECG | 10.54 | 24.27 | 0.67 | 5.46s | 66.48T |
| Ours | **7.21** | **21.63** | **0.66** | 0.85s | 5.91T |

reconstruction error from our 16 experts in the fusion MoE module. Interestingly, we find that the two methods share a similar principle, but our fusion MoE is much more efficient and requires only one inference. More details in Appendix C.

### 4.3 DISCUSSIONS

**Channel-wise Reconstruction Error Distribution.** To validate our original motivation for incorporating MoE model design, we visualized the cumulative errors of each channel for both our method and the DeScoD-ECG model using a histogram. From Fig. 5, we observe that the single model fails to maintain stable performance across different channels within the same sample, and it also struggles to achieve consistent performance for the same channel across different samples.

In contrast, our MoE-based method allows different channels to select the most suitable expert models, ensuring stable performance not only across different channels within the same sample but also for the same channel across different samples. Furthermore, in terms of cumulative error, our method significantly outperforms the single model DeScoD-ECG across channels. Besides, we also conduct an ablation study to evaluate the individual contributions of both the proposed Fusion MoE and RFAMoE modules to the overall performance (more details in D.1 of Appendix D).

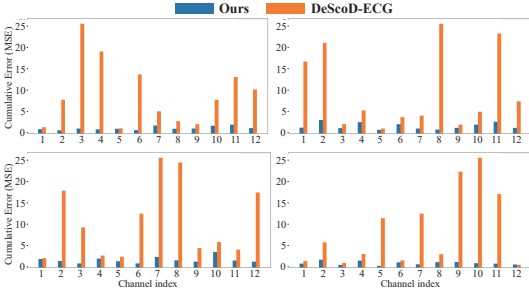

Figure 5: Comparison of the reconstruction across different channels between our method and the SOTA method DeScoD-ECG on four different data samples. Each figure represents a single 12-channel ECG sample, where the x-axis denotes the number of channels, and the y-axis represents the reconstruction quality indicated by cumulative MSE error (the lower, the better).

**Limitations.** In this work, we mainly focus on exploring the diffusion-based reconstruction and imputation of physiological time series data. Our proposed methods are supposed to be generalizable to other types of data and tasks, such as general time series data and tasks. It is worth further exploring the effectiveness of our methods in those fields in the future.

## 5 CONCLUSION

In this work, we present a conditional Diffusion-based MoE framework specifically designed for medical time series imputation. The proposed method introduces a RFAMoE block, providing the model with enhanced capacity to handle the highly variant periodicity among channels in medical time series signals caused by device variations and individual physiological differences. For the first time in diffusion-based time series tasks, we deploy a channel-level fusion MoE to assist in generating fused noise in a single forward pass. Our findings reveal that this approach exhibits a similar trend to K-shot averaging on improving reconstruction accuracy, and achieving better performance than K-shot averaging while maintaining a greater efficiency. We hypothesize that this method may emerge as a general and efficient approach for enhancing accuracy in diffusion-based time series reconstruction tasks.

## CLAIM OF LLM

In this work, large language models (LLMs) were used solely as a general-purpose writing assistant. Their role was limited to correcting grammar, fixing typographical errors, and polishing the language for clarity and readability.

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

# Appendix

## A    EVALUATION DATASETS

**PTB-XL dataset Wagner et al. (2020):**    The PTB-XL dataset is a large publicly available ECG dataset containing 12-channel recordings from 18,869 subjects, labeled with five diagnostic categories (four heart disease types and one healthy control). Each subject may have multiple trials, but only subjects with consistent diagnoses across trials are included, reducing the dataset to 17,596 subjects. The dataset offers recordings in both 100Hz and 500Hz versions, and for this study, we use the 100Hz version. After applying standard scaling, each trial (each 10 seconds long) is considered as one individual samples. The dataset is split using a subject-independent approach, with 60% of subjects assigned to the training set, 20% to the validation set, and 20% to the test set.

**SleepEDF dataset Kemp et al. (2000):**    The SleepEDF dataset includes 153 full-night polysomnographic (PSG) sleep recordings, encompassing EEG, EOG, chin EMG, and event markers. These recordings were collected from a study conducted between 1987 and 1991 to investigate age-related effects on sleep in healthy Caucasians aged 25 to 101 years. No sleep-related medications were administered, ensuring that the data captures natural sleep patterns. The sleep stages are annotated in 30-second epochs following the Rechtschaffen and Kales (R&K) standard (Wolpert, 1969), categorizing stages as Wake (W), REM, and Non-REM (N1, N2, N3). For this study, we select the [signal] channels and segment the data into 3000-point windows corresponding to the annotated sleep stages. A subject-wise split is employed, with 60% of subjects allocated to training, 20% to validation, and 20% to testing.

## B    MORE COMPARISON RESULTS OF IMPUTATION TASK

### B.1    COMPARISON WITH MoE METHODS

To complement the experiment results, we further provide additional comparisons with two open sourced MoE based methods (Time-MoE Shi et al. (2024) and Residual MoE Lee & Hauskrecht (2022)) on PTB imputation task. The results shown in Table B.1 show that our method significantly exceeds the Time-MoE Shi et al. (2024) and Residual MoE Lee & Hauskrecht (2022) on PRD and SSD metrics. In addition, our method is also leading in the MAD metric, compared with Time-MoE and Residual MoE.

### B.2    PTB IMPUTATION

To better illustrate the performance of each approach, and as a supplement to the results and discussion in Section 4.1, we provide visualizations of the imputed signals imputed by different methods.

In the figures below, the orange curves represent the imputed signals imputed by the models, while the blue curves correspond to the ground truth. The gray shaded regions indicate the missing parts of the input. Since the task is imputation, we primarily focus on how well the model reconstructs the missing segments (gray regions).

The imputation visualizations clearly demonstrate that our method outperforms all baseline methods. Notably, it surpasses DeScoD-ECG by providing more precise predictions, especially at critical points such as signal peaks and troughs (observed on (a) and (b) in Fig B.2). Among the baseline methods, TimesNet achieves the best performance. The trend is also observed on other data samples in the dataset.

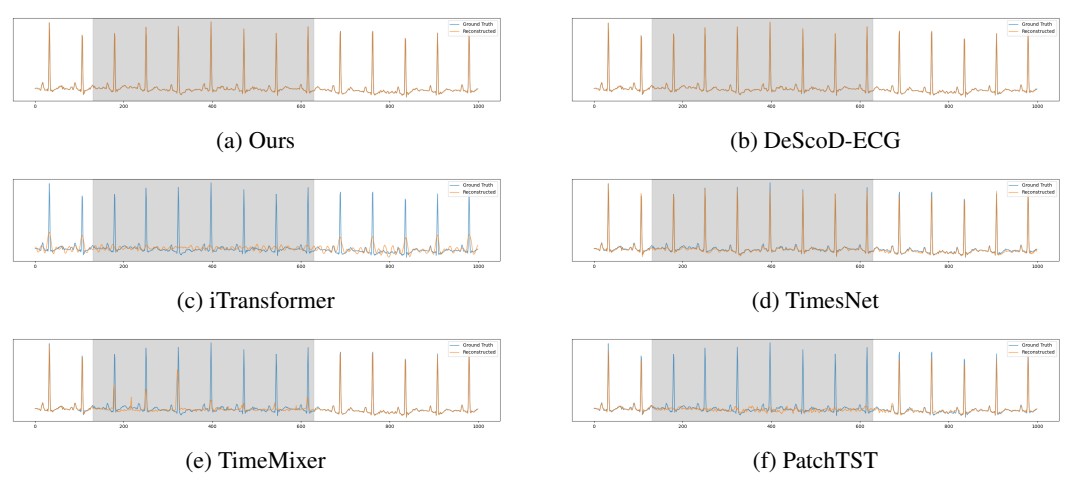

(a) Ours                                             (b) DeScoD-ECG

(c) iTransformer                                     (d) TimesNet

(e) TimeMixer                                        (f) PatchTST

Figure B.1: Imputation results under drop length=500 and drop channel=1.

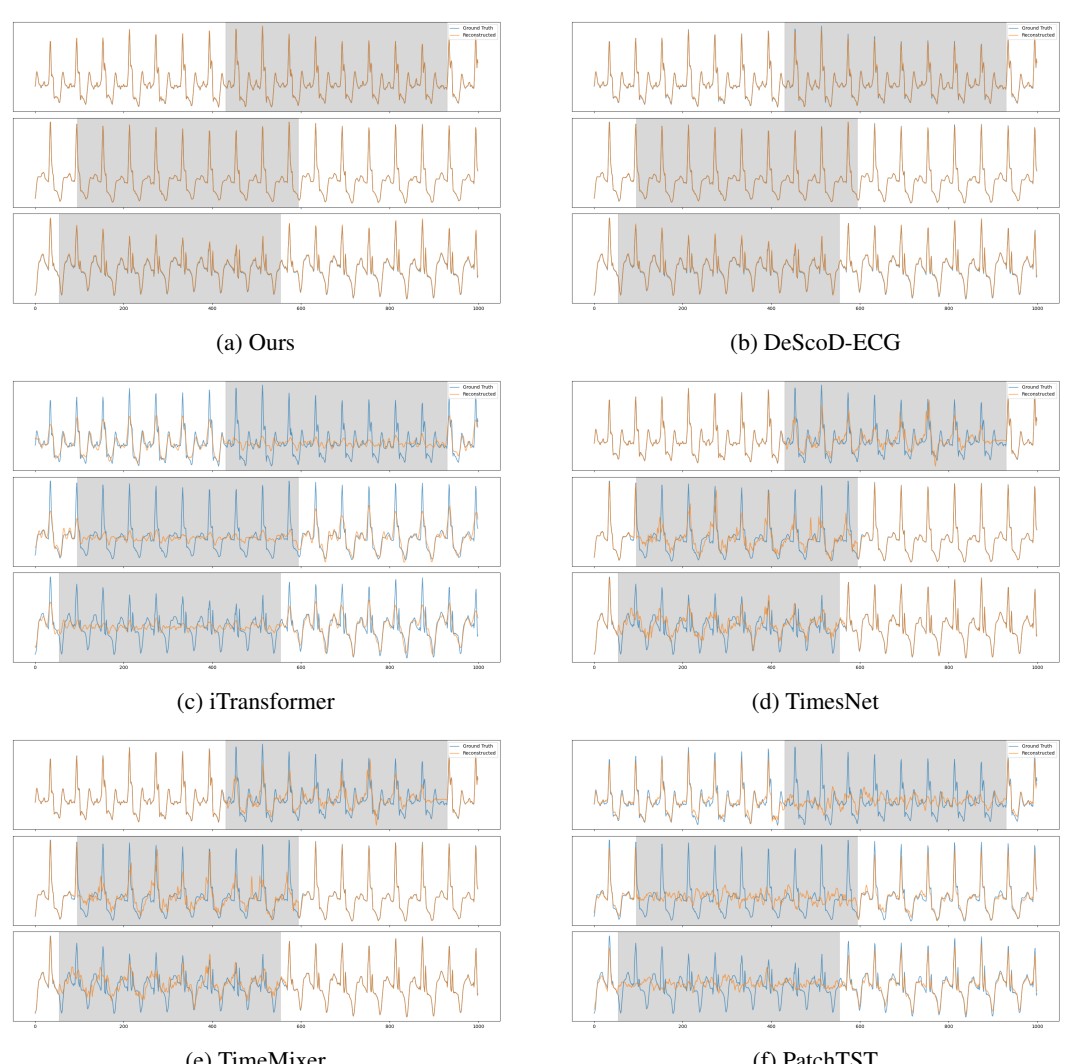

(a) Ours                                             (b) DeScoD-ECG

(c) iTransformer                                     (d) TimesNet

(e) TimeMixer                                        (f) PatchTST

Figure B.2: Imputation results under drop length=500 and drop channels=3.

Table B.1: Comparison results of MoE based methods on the imputation task of PTB dataset. The 300/500/800 represents the length of the contiguously missed data from different numbers of channels (i.e., 1,3,5,7,9, and 12).

| **Models** | | Ours | | | Time-MoE | | | Residual MoE | | |
|---|---|---|---|---|---|---|---|---|---|---|
| Metrics | | PRD | MAD | SSD | PRD | MAD | SSD | PRD | MAD | SSD |
| Drop 1 Channel | 300 | **4.16** | **0.01** | **0.48** | 101.94 | 0.37 | 292.02 | 106.58 | 1.89 | 321.44 |
| | 500 | **4.34** | **0.01** | **0.86** | 110.71 | 0.40 | 552.81 | 115.54 | 1.90 | 586.92 |
| | 800 | **11.73** | **0.04** | **10.45** | 119.62 | 0.42 | 1098.44 | 124.07 | 1.93 | 1126.74 |
| Drop 3 Channels | 300 | **5.86** | **0.06** | **1.86** | 103.44 | 1.06 | 298.98 | 108.24 | 2.66 | 327.89 |
| | 500 | **4.65** | **0.02** | **1.53** | 113.30 | 1.16 | 639.20 | 118.41 | 2.97 | 664.52 |
| | 800 | **14.35** | **0.17** | **20.35** | 125.36 | 1.22 | 1281.61 | 129.73 | 3.03 | 1309.17 |
| Drop 5 Channels | 300 | **5.85** | **0.06** | **1.86** | 103.62 | 1.79 | 293.98 | 109.12 | 3.48 | 326.83 |
| | 500 | **19.51** | **0.36** | **18.07** | 113.18 | 1.94 | 613.18 | 117.78 | 3.83 | 647.54 |
| | 800 | **61.73** | **1.28** | **264.73** | 124.40 | 2.05 | 1225.46 | 129.21 | 3.94 | 1255.09 |
| Drop 7 Channels | 300 | **12.36** | **0.24** | **12.54** | 103.95 | 2.44 | 318.51 | 109.59 | 4.32 | 347.76 |
| | 500 | **40.02** | **1.21** | **83.25** | 113.73 | 2.66 | 649.17 | 118.49 | 4.49 | 683.40 |
| | 800 | **83.53** | **2.43** | **514.63** | 124.88 | 2.82 | 1313.14 | 129.71 | 4.54 | 1342.60 |
| Drop 9 Channels | 300 | **14.41** | **0.35** | **13.40** | 102.61 | 3.13 | 317.39 | 106.95 | 4.84 | 346.98 |
| | 500 | **37.91** | **1.48** | **84.26** | 112.56 | 3.40 | 649.95 | 117.05 | 5.09 | 680.52 |
| | 800 | **84.02** | **3.13** | **533.00** | 123.44 | 3.60 | 1325.28 | 128.01 | 5.38 | 1354.41 |
| Drop 12 Channels | 300 | **17.39** | **0.57** | **27.45** | 103.29 | 4.27 | 327.09 | 109.01 | 6.10 | 357.90 |
| | 500 | **55.88** | **3.87** | **159.95** | 112.66 | 4.63 | 699.11 | 117.23 | 6.19 | 733.49 |
| | 800 | **95.21** | **4.63** | **643.14** | 123.22 | 4.88 | 1434.14 | 128.33 | 6.66 | 1466.83 |

## B.3 SLEEP EDF IMPUTATION

In addition, we visualize imputation results of models on the Sleep-EDF dataset. From these visualizations, we observe that the baseline methods are almost completely invalid, and we found that the counter-intuitive point is that when the model fails and does not predict anything (e.g., iTransformer, PatchTST, TimesNet, TimeMixer), it can get better metrics on the PRD. In both Figs B.3, and Fig B.4, iTransformer, PatchTST are invalid on predicting, the results are close to a smooth straight line. This trend was observed on other data samples. However, these two methods get better performance on PRD than other methods in table B.2. In Figs B.3, and B.4, TimesNet and TimeMixer also exhibit near failure under this setting, they still preserve little predictive capability, which can be also observed on other data samples in the dataset. And both of them achieve a worse performance in evaluation than iTransformer and PatchTST. Although our method and DeScoD-ECG did not achieve strong predictive performance, they did not completely lose their ability to make predictions. Their performance on the evaluation metrics was comparable to that of TimesNet and TimeMixer, albeit still worse than iTransformer and PatchTST.

We conjecture that the failure of all baseline methods on the Sleep-EDF dataset is due to the near-zero correlation between channels, as the signals are collected from different sensors and distinct parts of the body. Interestingly, we observe that when a model completely refrains from making predictions, it may not yield the worst evaluation scores. This phenomenon is also observed in reconstruction tasks on Sleep-EDF (baseline methods are invalid on reconstructing signals without achieving the bad performance on evaluation metrics.) This also raises a new concern: the need for more reliable evaluation metrics to prevent such cases.

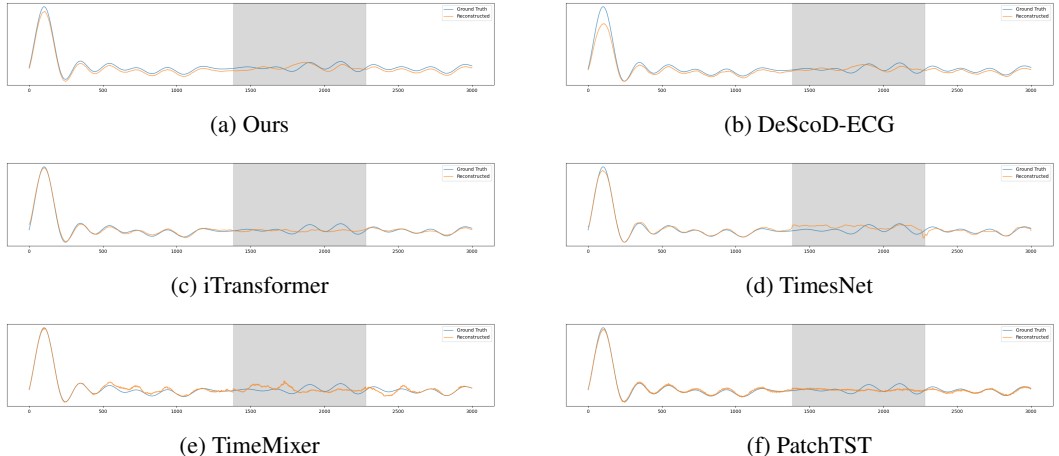

Figure B.3: Imputation results under drop length=900 and drop channels=1.

Table B.2: Comparison results of the imputation task on Sleep EDF dataset. The 900/1200/1500/2000 represents the length of the contiguously missed data from different numbers of channels (i.e., 1,3 and 6)

| Models Metrics | | Ours | | | DeScoD-ECG | | | iTransformer | | | TimesNet | | | TimeMixer | | | PatchTsT | | |
|---|---|---|---|---|---|---|---|---|---|---|---|---|---|---|---|---|---|---|---|---|
| | | PRD | MAD | SSD | PRD | MAD | SSD | PRD | MAD | SSD | PRD | MAD | SSD | PRD | MAD | SSD | PRD | MAD | SSD |
| Drop 1 Channel | 900 | 108.94 | 0.27 | 600.19 | 115.17 | 0.31 | 656.50 | 96.88 | 0.25 | 519.46 | 116.47 | 0.29 | 715.11 | 109.79 | 0.28 | 618.71 | 102.93 | 0.26 | 547.69 |
| | 1200 | 109.68 | 0.30 | 913.10 | 115.33 | 0.33 | 988.90 | 97.69 | 0.28 | 781.19 | 117.12 | 0.31 | 1028.55 | 110.52 | 0.30 | 907.94 | 105.29 | 0.29 | 836.90 |
| | 1500 | 111.64 | 0.33 | 1265.22 | 116.13 | 0.34 | 1331.38 | 98.25 | 0.30 | 1074.21 | 118.42 | 0.33 | 1450.83 | 112.48 | 0.32 | 1306.24 | 107.56 | 0.31 | 1162.29 |
| | 2000 | 109.40 | 0.35 | 1912.57 | 111.22 | 0.36 | 1955.65 | 98.84 | 0.33 | 1698.03 | 121.33 | 0.36 | 2944.58 | 116.89 | 0.36 | 2486.53 | 112.13 | 0.35 | 2098.32 |
| Drop 3 Channel | 900 | 108.37 | 0.86 | 830.26 | 115.58 | 0.91 | 882.49 | 96.75 | 0.78 | 744.39 | 116.24 | 0.87 | 947.59 | 109.58 | 0.87 | 829.27 | 102.44 | 0.80 | 742.29 |
| | 1200 | 109.88 | 0.94 | 1030.28 | 116.63 | 0.98 | 1100.39 | 97.73 | 0.85 | 893.74 | 117.52 | 0.94 | 1207.07 | 111.04 | 0.93 | 1008.35 | 105.38 | 0.88 | 858.83 |
| | 1500 | 111.94 | 1.01 | 1256.31 | 116.72 | 1.06 | 1320.54 | 98.30 | 0.90 | 1056.54 | 119.37 | 1.00 | 1564.69 | 112.46 | 0.98 | 1242.45 | 107.74 | 0.94 | 1148.19 |
| | 2000 | 109.44 | 1.09 | 1969.02 | 111.45 | 1.12 | 2009.82 | 98.78 | 0.99 | 1724.62 | 119.65 | 1.09 | 2368.25 | 115.51 | 1.08 | 2093.06 | 111.05 | 1.05 | 1845.16 |
| Drop 6 Channel | 900 | 115.20 | 2.62 | 957.15 | 119.50 | 2.70 | 1001.28 | 98.04 | 2.27 | 778.01 | 108.49 | 2.37 | 882.45 | 104.91 | 2.36 | 814.34 | 101.49 | 2.30 | 750.12 |
| | 1200 | 116.01 | 2.85 | 1304.58 | 119.31 | 2.91 | 1351.14 | 98.58 | 2.48 | 1038.90 | 109.05 | 2.57 | 1181.19 | 105.52 | 2.56 | 1098.71 | 102.73 | 2.51 | 1036.82 |
| | 1500 | 116.60 | 2.98 | 1646.59 | 118.62 | 3.03 | 1684.37 | 98.87 | 2.60 | 1296.59 | 110.01 | 2.71 | 1537.86 | 106.27 | 2.69 | 1394.78 | 103.78 | 2.65 | 1321.37 |
| | 2000 | 110.72 | 3.09 | 2011.19 | 115.27 | 3.20 | 2142.22 | 99.19 | 2.83 | 1704.90 | 110.24 | 2.93 | 2157.60 | 107.98 | 2.93 | 1973.19 | 105.49 | 2.89 | 1830.17 |

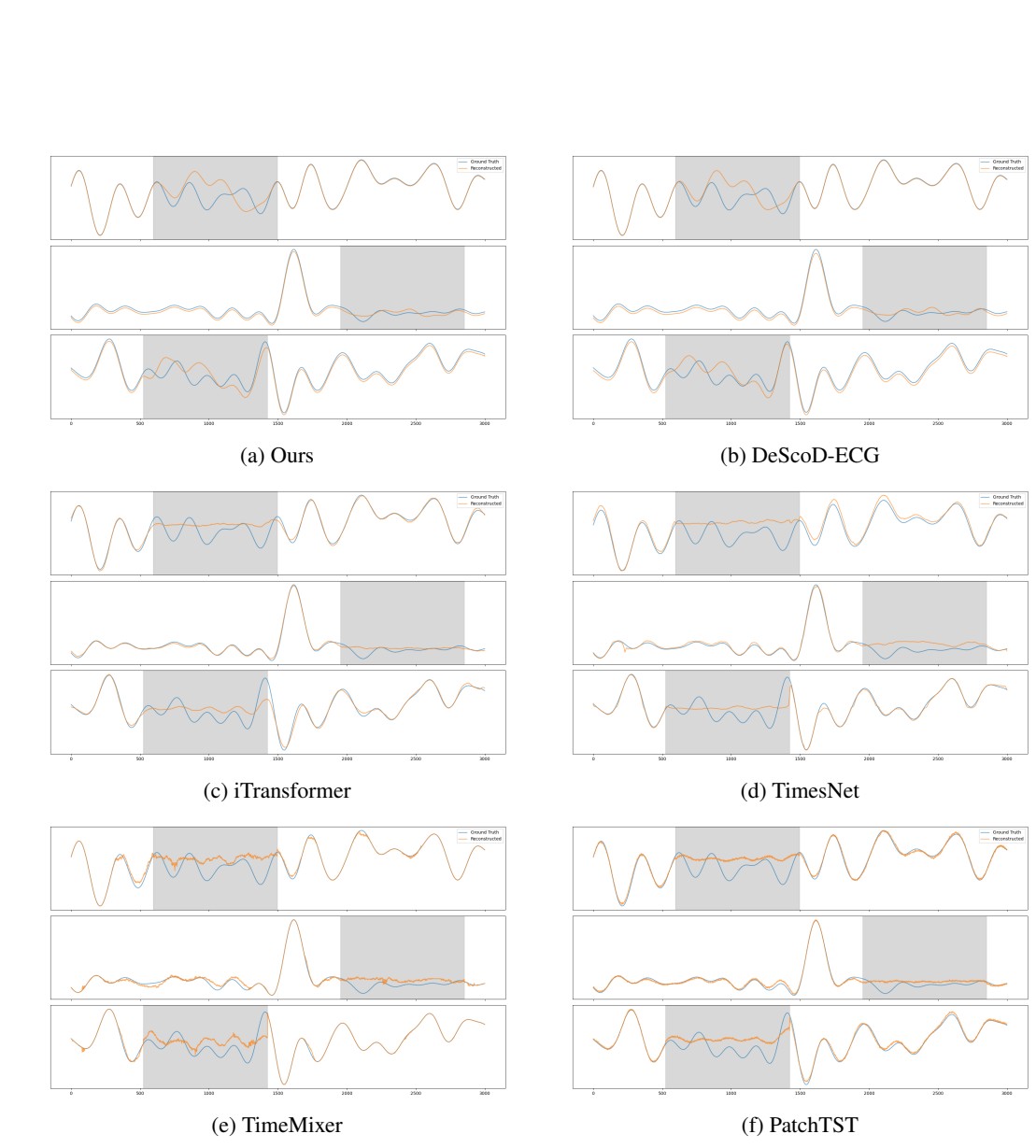

Figure B.4: Imputation results under drop length=900 and drop channels=3.

## C    Exploring the Similarity Between Fusion MoE and K-Shot Averaging

To further explore the rationale behind this superior performance, we observe the distribution of reconstruction error from DeScoD-ECG over 12 shots and compare it with the distribution of reconstruction error from our 12 experts in the fusion MoE module. Interestingly, we find that the two methods share a similar principle, but our fusion MoE is much more efficient and requires only one inference.

We sequentially fix the final layer of the network to each expert in the well trained Fusion MoE model, resulting in 12 different models. These models are identical except for the convolutional layer at the output of the backbone, where each model utilizes a different expert. We then use these 12 models to reconstruct the test dataset. We select a channel and plot the reconstruction errors (Fig. C.2) obtained from each of the 12 individual models on it, as well as the reconstruction error produced by the full Fusion MoE model.

Additionally, to further compare our approach with the 12-shot averaging method. In Fig. C.1, we visualize the reconstruction errors for each individual shot in the 12-shot setting and compare them with the error of the averaged reconstruction.

In the Figs C.2, and C.1, below, the x-axis represents the time stamps of a single channel (length = 1000), and the y-axis indicates the reconstruction error at each time stamps.

Here we formulate the model's single one-shot reconstruction output as:

$$\hat{Y} = Y + \Delta \tag{6}$$

where $\hat{Y}$ denotes the reconstructed signal, $Y$ is the ground truth, and $\Delta$ represents the reconstruction error by this shot. For each of the 12 independent shots, the DeSco-ECG generates a distinct set of reconstruction errors across all time stamps of the selected channel, resulting in 12 different $\Delta$ ($\Delta_1$, $\Delta_2$, $\Delta_3$, $\Delta_4$ .... $\Delta_{12}$ ). Eventually, we obtain:

$$\hat{Y}_{avg} = [(Y + \Delta_1) + (Y + \Delta_2) + ....(Y + \Delta_{12})]/12 \tag{7}$$

In the Fig C.1, each group of identically colored points except red points corresponds to a $\Delta$ generated by one single shot. From the Fig C.1, we can find $\Delta_1$, $\Delta_2$, $\Delta_3$ ....$\Delta_{12}$ are roughly symmetrically distributed around the zero axis across all time stamps.(Observed on all data samples) As a result, averaging across multiple shot predictions, the $\Delta_{avg} = (\Delta_1 + \Delta_2 + ... \Delta_{12}) / 12$, which is represented by the group of red points (Average) in the Fig C.1, will be close to $0$.

In the Fig C.2, each group of identically colored points except red points corresponds to the reconstruction errors across all time stamps when the Fusion MoE only activate this corresponding expert reconstructing the signals. Unlike the Multi-shot Average, our methods allows each expert to generate a distinct denoise factor $\varepsilon$ at each time step, then according to the denoising formula from the diffusion model, the final reconstruction output $Y$ is computed as:

$$\hat{Y} = Noise - \sum_{t=1}^{T} \sigma_t \cdot \varepsilon_t$$

where $Noise \sim \mathcal{N}(0, I)$ is Gaussian noise, $\varepsilon_t$ is the denoise factor generated by the expert at each timestep, and $\sigma_t$ represents the noise scale controlling the magnitude of denoising at each step.

After obtaining $Y$, we follow the same procedure used in multi-shot averaging and denote it as:

$$\hat{Y} = Y + \Delta \tag{8}$$

where $\hat{Y}$ denotes the reconstructed signal, $Y$ is the ground truth, and $\Delta$ represents the reconstruction error by this expert.

When we reconstruct the signal using each expert individually, we obtain a set of expert-specific reconstructions denoted as:

$$\hat{Y}_1 = Noise - \sum_{t=1}^{T} \sigma_t \cdot \varepsilon_{1,t} = Y + \Delta_1$$

$$\hat{Y}_2 = Noise - \sum_{t=1}^{T} \sigma_t \cdot \varepsilon_{2,t} = Y + \Delta_2$$

$$\hat{Y}_3 = Noise - \sum_{t=1}^{T} \sigma_t \cdot \varepsilon_{3,t} = Y + \Delta_3$$

$$\vdots$$

$$\hat{Y}_{12} = Noise - \sum_{t=1}^{T} \sigma_t \cdot \varepsilon_{12,t} = Y + \Delta_{12}$$

Each of the $\Delta_1, \Delta_2, \Delta_3, \ldots, \Delta_{12}$ corresponds to a distinct group of identically colored points (except red points) in Fig C.2, and is representing reconstruction errors generated by $Expert_1$, $Expert_2$, $Expert_3$,... $Expert_{12}$ respectively.

And we find that $\Delta_1, \Delta_2, \Delta_3, \ldots, \Delta_{12}$ in this case also symmetrically distributed around the zero axis across all time stamps.

However, when we activate all experts in Fusion MoE, the reconstruction can be denoted as:

reflecting the reconstruction errors across all time steps for that shot.

$$\hat{Y}_{Fusion} = Noise - \sum_{t=1}^{T} \sigma_t \cdot (k_1 \varepsilon_{1,t} + k_2 \varepsilon_{2,t} + k_3 \varepsilon_{3,t} + \ldots + k_{12} \varepsilon_{12,t}) = Y + \Delta_{Fusion}$$

Where $k_1, k_2, \ldots, k_{12}$ are the gating weights assigned to each expert at each time step $t$, and $\sum_{i=1}^{12} k_i = 1$. It turns out that $\Delta_{Fusion}$ represented by the group of red points (Fusion Result) in Fig C.2, is close to 0. And such trend on Fusion MoE is also observed on other data samples

We observe that both methods exhibit a similar trend in reducing reconstruction errors, providing empirical evidence of their underlying similarity. This suggests that our method inherently achieves a similar effect to multi-shot averaging but with a more efficient computational design, making it a more practical choice for real-world applications. This further opens up promising directions for future research into how the MoE mechanism, by producing multiple denoising factors from the same noisy input and combining them through a gating mechanism, can emulate the effect of averaging reconstructions obtained from multiple independent denoising on different noisy inputs.

Furthermore, in Appendix E, we present a mathematical derivation demonstrating that our Fusion MoE framework yields lower reconstruction error compared to the Multi-Shot approach.

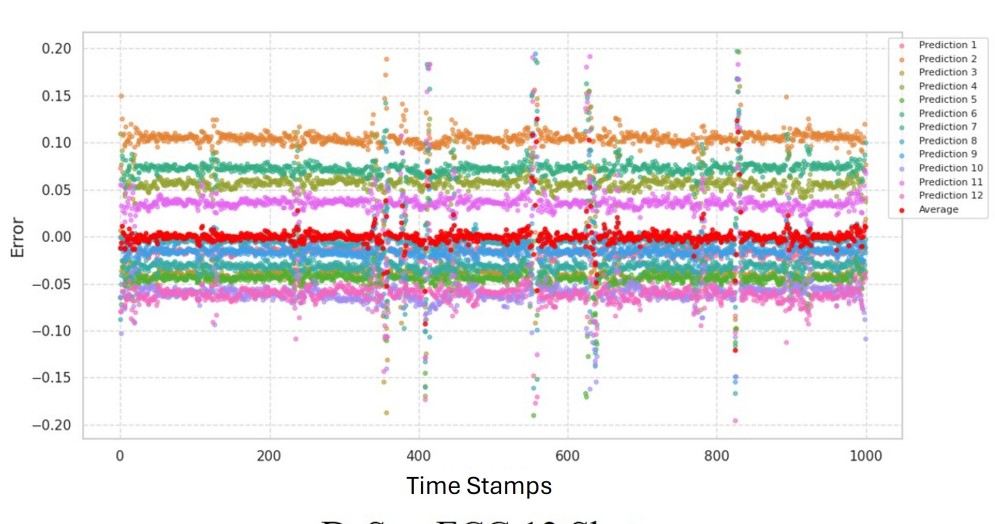

DeSco-ECG 12 Shot

Figure C.1: Error distribution for reconstructing a single channel 12 times with DeScoD-ECG; Red scatters indicate the averaged reconstruction error.

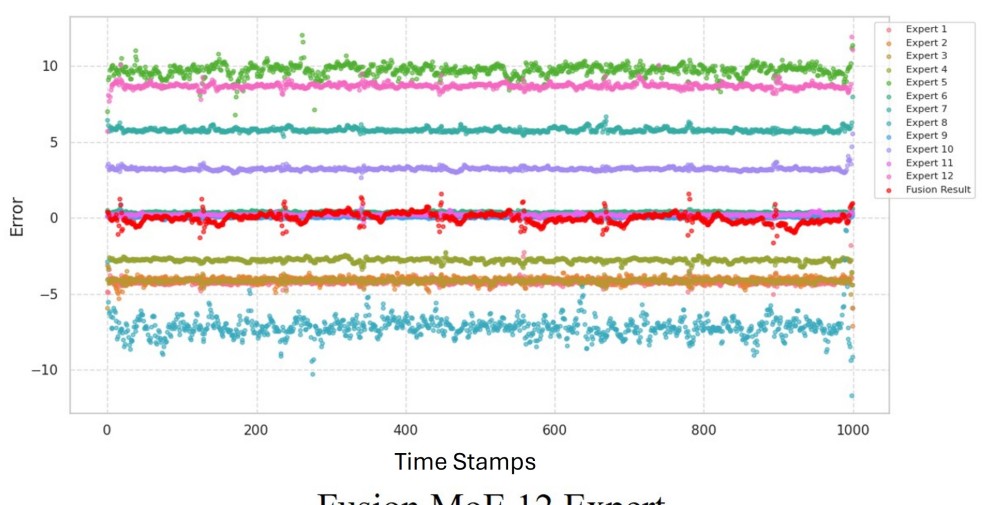

Fusion MoE 12 Expert

Figure C.2: Error distribution for reconstructing a single channel using fixed expert within Fusion MoE. Red scatters indicates errors of using fusion MoE.

## D ABLATION ON FUSION MOE AND RFAMOE

### D.1 DISENTANGLING THE CONTRIBUTIONS OF FUSION MOE AND RFAMOE

To disentangle the contributions of the Fusion MoE and the RFAMoE modules, we conduct ablation studies on the PTB reconstruction task by adding each component individually to the DeScoD-ECG.

Table D.1 reports the results of adding the Fusion MoE and RFAMoE modules individually on top of the DeScoD-ECG baseline. We observe that incorporating either Fusion MoE or RFAMoE alone already leads to significant improvements over the baseline, demonstrating the effectiveness of Fusion MoE and RFAMoE, respectively. More importantly, when both modules are combined, the model achieves further performance gains across all three metrics (PRD, SSD, and MAD). This indicates that the two modules are complementary. Their synergy ultimately drives the best overall reconstruction quality.

Table D.1: Comparison results between the models with and without Fusion MoE on the PTB dataset.

|  | PRD $\downarrow$ | SSD $\downarrow$ | MAD $\downarrow$ |
|---|---|---|---|
| DeScoD-ECG (Baseline) | 16.54 | 52.71 | 0.98 |
| + Only Fusion MoE | 10.32 | 31.82 | 0.81 |
| + Only RFAMoE | 9.73 | 28.58 | 0.73 |
| Fusion MoE + RFAMoE | 7.21 | 21.36 | 0.66 |

### D.2 HOW TO SELECT K INSIDE THE FUSION MOE

To further investigate the effect of the number of activated experts $K$ in the Fusion MoE, we perform the ablation study on the PTB imputation task. Specifically, we adopt the setting where random 300 consecutive time steps are dropped from 5 channels, which provides a challenging scenario to evaluate different values of $K$. Table D.2 summarizes the results.

As shown, increasing $K$ consistently improves the reconstruction performance across PRD, MAD, and SSD when moving from $K = 4$ to $K = 16$, demonstrating that activating more experts enables increase of the performance. However, the improvement becomes marginal once $K$ exceeds 16, with PRD, MAD, and SSD remaining nearly unchanged from $K = 16$ to $K = 32$. This indicates a diminishing return, where additional experts no longer bring substantial accuracy gains.

Table D.2: Ablation study on the number of activated experts $K$ in the Fusion MoE, conducted on the PTB imputation task (drop 300-length on 5 channels). Performance generally improves as $K$ increases, with diminishing returns observed after $K = 16$.

| $K$ | 4 | 8 | 12 | 16 | 20 | 24 | 28 | 32 |
|---|---|---|---|---|---|---|---|---|
| PRD | 14.83 | 13.93 | 11.56 | 5.90 | 5.85 | 5.69 | 5.55 | 5.55 |
| MAD | 0.078 | 0.074 | 0.070 | 0.061 | 0.060 | 0.059 | 0.063 | 0.063 |
| SSD | 4.78 | 4.64 | 4.42 | 1.781 | 1.73 | 1.72 | 1.57 | 1.57 |

## E PROOF

Here, we provide a five-step mathematical proof to show that the performance of Fusion MoE is no worse than that of the Multi-Shot Average.

*Proof.* We structure the proof in five steps.

**Step 1: Probability Distributions** $p\big(x_{t-1} \mid g_\theta\big)$ **and** $p\big(g_\theta \mid x_t, t, \tilde{x}\big)$**.** In typical diffusion models:

$$x_{t-1} = \epsilon_D(x_t, g_\theta, t) = A(t)\, x_t + B(t)\, g_\theta. \tag{9}$$

Hence, conditionally on $g_\theta$ (and fixed $x_t, t$), $x_{t-1}$ is often a *deterministic* function of $g_\theta$. Symbolically,

$$p\big(x_{t-1} \mid g_\theta\big) = \delta\Big(x_{t-1} - \epsilon_D(x_t,\, g_\theta,\, t)\Big), \qquad (10)$$

a Dirac delta at $\epsilon_D(\cdot, \cdot, \cdot)$.

We also have a (possibly stochastic) *noise-prediction* model $p\big(g_\theta \mid x_t, t, \tilde{x}\big)$ that describes how each $g_\theta$ is drawn (e.g., from a set of $K$ experts or from a mixture of them).

**Step 2: Expected Loss in the General Setting.**    Given $L(\cdot)$ is a loss function on $x_{t-1} \in \mathbb{R}^d$, the *expected* loss is

$$\mathbb{E}\big[L(x_{t-1})\big]$$
$$= \iint L(x_{t-1})\, p\big(x_{t-1} \mid g_\theta\big)\, p\big(g_\theta \mid x_t,\, t,\, \tilde{x}\big)\, dx_{t-1}\, dg_\theta. \qquad (11)$$

But since $x_{t-1} = \epsilon_D(x_t, g_\theta, t)$ is deterministic given $g_\theta$ (for each $t$),

$$\mathbb{E}\big[L(x_{t-1})\big] = \int L\big(\epsilon_D(x_t, g_\theta, t)\big)\, p\big(g_\theta \mid x_t,\, t,\, \tilde{x}\big)\, dg_\theta. \qquad (12)$$

Thus, analyzing $\epsilon_D$ *as a function of* $g_\theta$ suffices to compare different denoising strategies.

**Step 3: $K$-Run + Averaging.**    If we run each expert $k$ separately and then average, we define expert $k$ as $g_\theta^{(k)}(x_t, t, \tilde{x})$:

$$x_{t-1}^{(k)} = \epsilon_D\big(x_t,\, g_\theta^{(k)},\, t\big) = A(t)\, x_t + B(t)\, g_\theta^{(k)}. \qquad (13)$$

Finally,

$$\overline{x}_{t-1} = \frac{1}{K} \sum_{k=1}^{K} x_{t-1}^{(k)} = A(t)\, x_t + B(t) \left( \frac{1}{K} \sum_{k=1}^{K} g_\theta^{(k)} \right). \qquad (14)$$

**Step 4: Single-Step Mixture-of-Experts (MoE).**    MoE combines the experts' noise estimates *first*, via nonnegative weights $w^{(k)}(\tilde{x})$ summing to 1:

$$\hat{\epsilon}_{\text{MoE}} = \sum_{k=1}^{K} w^{(k)}(\tilde{x})\, g_\theta^{(k)}(x_t, t, \tilde{x}). \qquad (15)$$

Because $\epsilon_D$ is linear in its second argument,

$$x_{t-1}^{\text{MoE}} = \epsilon_D\big(x_t,\, \hat{\epsilon}_{\text{MoE}},\, t\big)$$
$$= A(t)\, x_t + B(t) \sum_{k=1}^{K} w^{(k)}\, g_\theta^{(k)}$$
$$= \sum_{k=1}^{K} w^{(k)} \big( A(t)\, x_t + B(t)\, g_\theta^{(k)} \big) \qquad (16)$$
$$= \sum_{k=1}^{K} w^{(k)}\, x_{t-1}^{(k)}.$$

Hence $x_{t-1}^{\text{MoE}}$ is a *convex combination* of $\{x_{t-1}^{(k)}\}_{k=1}^{K}$.

**Step 5: Jensen's Inequality and the Expected Loss Comparison.**    Since $L(\cdot)$ is **convex**, we have for any convex combination (weights $\alpha_k \geq 0$, $\sum_k \alpha_k = 1$):

$$L\Big(\sum_k \alpha_k\, y_k\Big) \;\leq\; \sum_k \alpha_k\, L(y_k). \qquad (17)$$

Applying this to $y_k = x_{t-1}^{(k)}$ and $\alpha_k = w^{(k)}(\tilde{x})$, we get

$$L\big(x_{t-1}^{\text{MoE}}\big) = L\Big(\sum_{k=1}^{K} w^{(k)}\, x_{t-1}^{(k)}\Big) \;\leq\; \sum_{k=1}^{K} w^{(k)}\, L\big(x_{t-1}^{(k)}\big). \qquad (18)$$

Choosing $w^{(k)} = \frac{1}{K}$ recovers $\overline{x}_{t-1}$, so

$$L\left(\overline{x}_{t-1}\right) = L\left(\frac{1}{K}\sum_{k=1}^{K} x_{t-1}^{(k)}\right) \leq \frac{1}{K}\sum_{k=1}^{K} L\left(x_{t-1}^{(k)}\right). \tag{19}$$

Since MoE can *choose any weights* in the simplex, it cannot perform worse than uniform averaging:

$$L\left(x_{t-1}^{\mathrm{MoE}}\right) \leq L\left(\overline{x}_{t-1}\right). \tag{20}$$

Taking the expectation over $g_\theta \sim p\left(g_\theta \mid x_t,\, t,\, \tilde{x}\right)$ (and any other randomness), we use linearity of expectation plus the fact that $x_{t-1}$ is deterministically linked to $g_\theta$. Thus

$$\mathbb{E}\left[L\left(x_{t-1}^{\mathrm{MoE}}\right)\right] \leq \mathbb{E}\left[L\left(\overline{x}_{t-1}\right)\right]. \tag{21}$$

Hence, a single-step MoE approach is **not worse** than a $K$-expert separate-run averaging strategy. This completes the proof.

$\square$

