# OpenReview forum: "Advancing Physiological Time Series Reconstruction and Imputation via Mixture of Receptive Fields and Experts Fusion"
_ICLR.cc/2026/Conference — ICLR 2026 Conference Withdrawn Submission_

### Official Review · Reviewer_Xj4r · 2025-10-16

**Soundness:** 2
**Presentation:** 2
**Contribution:** 2
**Rating:** 2
**Confidence:** 4

**Summary:**

This paper addresses the problem of reconstruction and imputation for medical time series by proposing a Mixture of Experts (MoE)-based noise estimator within a score-based diffusion framework. Specifically, the method introduces a Receptive Field Adaptive MoE (RFAMoE) module and a Fusion MoE module. Extensive experiments are conducted to demonstrate the effectiveness and efficiency of the proposed approach.

**Strengths:**

- S1: This paper combines the Mixture of Experts (MoE) framework with score-based diffusion models.
- S2：The proposed Fusion MoE module presents an interesting idea to alleviate the high computational complexity issue faced in K-shot reconstruction.

**Weaknesses:**

- W1: The paper lacks comparisons with other diffusion-based models such as CSDI[1], MTSCI[2], FGTI[3], and Diffputer[4]. These methods are also omitted from the related work section, indicating that the authors need to provide a more comprehensive discussion of related studies and include stronger baseline experiments.
- W2：The ablation study introduces the RFAMoE and Fusion MoE modules on top of the DeScoD-ECG baseline to verify their effectiveness. However, there is no ablation experiment for the bridge module, which should be included to support the analysis.
- W3：As indicated around line 298, the proposed method builds upon DeScoD-ECG by incorporating RFAMoE and Fusion MoE for experimentation. Since the paper emphasizes that these modules are general and claims to propose a new framework, it would be important to evaluate the approach on other diffusion-based backbones as well. Currently, experiments are conducted on only a single backbone.
- W4：The experimental analysis is insufficient. According to Table 2, TimeMixer significantly outperforms the proposed method, yet the paper does not provide an explanation for this result.
- W5：The paper lacks a clear summary of its contributions and a precise description of the problem definition. Moreover, there are typographical issues in important figures, such as the word “out” being misspelled as “our” in Figure 3.
- W6：The code is not provided for reproducibility.

[1] CSDI: Conditional Score-based Diffusion Models for Probabilistic Time Series Imputation. NeurIPS 2021.
[2] MTSCI: A Conditional Diffusion Model for Multivariate Time Series Consistent Imputation. CIKM 2024.
[3] Frequency-aware Generative Models for Multivariate Time-series Imputation. NeurIPS 2024.
[4] DiffPuter: An EM-Driven Diffusion Model for Missing Data Imputation. ICLR 2025.

**Questions:**

- Q1: How are the three evaluation metrics computed? Why does the paper not use commonly adopted metrics in reconstruction and imputation tasks, such as MAE, RMSE, or MAPE?
- Q2：What is the difference between the reconstruction and imputation tasks in the experimental setup, and how is this distinction reflected in the configuration of the experiments?

---

### Official Review · Reviewer_koUw · 2025-10-26

**Soundness:** 3
**Presentation:** 2
**Contribution:** 2
**Rating:** 4
**Confidence:** 5

**Summary:**

The authors propose a novel Mixture of Experts (MoE)-based noise estimator within a score-based diffusion framework. The authors design a Fusion MoE module that leverages the inherent parallelism of the MoE structure to generate K noise signals simultaneously, fuse them through a routing mechanism, and complete signal reconstruction in a single inference step.  Extensive experimental results demonstrate that the proposed framework consistently outperforms existing diffusion-based state-of-the-art approaches across various tasks and datasets.

**Strengths:**

1.The idea of applying the Mixture of Experts (MoE) mechanism to diffusion models is highly novel and effectively reduces the computational and temporal costs associated with repeatedly executing forward processes.

2.The proposed RFAMoE module effectively integrates receptive fields of different sizes, enabling multi-scale modeling capabilities.

**Weaknesses:**

1. The experiments are conducted on relatively limited datasets (PTB and PSG), with a small number of baselines for comparison. Moreover, the non-diffusion baselines used are not specifically designed for imputation or reconstruction tasks. In addition, the authors should provide a clear explanation of the three evaluation metrics.

2. The introduction section lacks a concise summary of the main contributions of this work, and the writing needs further polishing for clarity and academic style.

**Questions:**

please refer to the weakness, and additionally:
1. Since the model is based on a score-based diffusion framework—where the sampling process inherently draws from the learned distribution—the authors should clarify how the MoE modeling approach influences the posterior distribution of the diffusion process.
2. I am doubtful about some of the parameter choices, particularly the kernel size used in the RFAMoE module. A parameter study might be necessary to justify these settings. In my opinion, selecting convolutional kernels with sizes ranging from 1 to 57 seems excessively large.

---

### Official Review · Reviewer_ufFJ · 2025-10-30

**Soundness:** 2
**Presentation:** 3
**Contribution:** 2
**Rating:** 2
**Confidence:** 5

**Summary:**

The authors propose a score-based diffusion method with a Mixture of Experts (MoE)-based noise estimator for time series reconstruction. The Receptive Field Adaptive MoE (RFAMoE) module is to adaptively select desired receptive fields for each channel during the diffusion process. The Fusion MoE module fuses the noise signals generated in parallel to complete signal reconstruction in a single inference step by using a routing mechanism.

**Strengths:**

1)	Using a mixture of expert models to improve the reconstruction performance for time series data seems feasible.
2)	The experimental results on different real-world time series datasets are given.
3)	Relevant ablation study experimental results are provided to prove the effectiveness of the designed modules.

**Weaknesses:**

1)	The authors claim that the MoE is no worse than the k-run averaging. However, the k-run averaging can use different noisy inputs for each run, which can result in better sampling diversity. while the input of the MoE is the same.
2)	In Eq (14) the drift term is A(t)x_t, but what if the drift term A(x_t,t) is nonlinear?
3)	Using MoE to improve the model performance is a relative naïve idea, therefore the novelty of the model is somewhat limited.
4)	The diversity of the outputs computed the experts can be limited as they share some building blocks.
5)	How to learn the weight for each expert is unclear.
6)	How to compute PRD, MAD, SSD exactly? And what are the difference compared to MSE and MAE?

**Questions:**

The proposed method is only tested on the physiological data. Is this method can be applied to other time series data types?

---

### Note · Authors · 2025-11-17

I have read and agree with the venue's withdrawal policy on behalf of myself and my co-authors.